# Beneficial Effects of Theaflavins on Metabolic Syndrome: From Molecular Evidence to Gut Microbiome

**DOI:** 10.3390/ijms23147595

**Published:** 2022-07-08

**Authors:** Meng Shi, Yuting Lu, Junling Wu, Zhibing Zheng, Chenghao Lv, Jianhui Ye, Si Qin, Chaoxi Zeng

**Affiliations:** 1Laboratory of Food Function and Nutrigenomics, College of Food Science and Technology, Hunan Agricultural University, Changsha 410128, China; shimeng@hunau.edu.cn (M.S.); luyuting1214@163.com (Y.L.); wujunling1114@163.com (J.W.); zhengzb712@sina.com (Z.Z.); lvchenghao0514@163.com (C.L.); 2Tea Research Institute, Zhejiang University, Hangzhou 310058, China; jx0515@163.com

**Keywords:** TFs, metabolic syndrome, mechanism, bioavailability, gut microbiota

## Abstract

In recent years, many natural foods and herbs rich in phytochemicals have been proposed as health supplements for patients with metabolic syndrome (MetS). Theaflavins (TFs) are a polyphenol hydroxyl substance with the structure of diphenol ketone, and they have the potential to prevent and treat a wide range of MetS. However, the stability and bioavailability of TFs are poor. TFs have the marvelous ability to alleviate MetS through antiobesity and lipid-lowering (AMPK-FoxO3A-MnSOD, PPAR, AMPK, PI3K/Akt), hypoglycemic (IRS-1/Akt/GLUT4, Ca^2+^/CaMKK2-AMPK, SGLT1), and uric-acid-lowering (XO, GLUT9, OAT) effects, and the modulation of the gut microbiota (increasing beneficial gut microbiota such as *Akkermansia* and *Prevotella*). This paper summarizes and updates the bioavailability of TFs, and the available signaling pathways and molecular evidence on the functionalities of TFs against metabolic abnormalities in vitro and in vivo, representing a promising opportunity to prevent MetS in the future with the utilization of TFs.

## 1. Introduction

Metabolic syndrome (MetS) presents a multiplex risk of atherosclerotic cardiovascular disease and type 2 diabetes, which globally occur commonly, with prevalence from 10 to 40%, leading to the requirement of controlling its risk factors [1]. MetS is defined by a cluster of interconnected factors that are marked by the presence of clusters of hyperglycemia, obesity, dyslipidemia, and hypertension [2], and it is usually linked to an increased risk of developing central obesity, insulin resistance, atherogenic dyslipidemia (high triglycerides, low HDL cholesterol) and hyperuricemia [3]. Its pathophysiology is primarily attributed to insulin resistance associated with excess fatty acid flux [4].

Phytochemical dietary interventions are gaining considerable attention as a prophylactic and therapeutic strategy due to their safety, better tolerability, and economic benefits. Dietary interventions with flavan-3-ols have beneficial effects on metabolic syndrome (MetS). Tea is the major single contributor (75%) to flavan-3-ol monomer intake, followed by pome fruits with 6%, while theaflavins are the main source of primary flavan-3-ol intake [5]. In countries where tea is consumed sparingly, flavonoid intake relies mainly on proanthocyanidins from berries, contributing to preventing MetS [6]. The average concentration of flavanols in mg per 100 g fresh weight (mg per 100 mL) is 58% for blueberries, 39% for red wine, and 28% for black tea [7]. The dietary intake of flavan-3-ol monomer, proanthocyanidins, and theaflavin varies significantly among countries, and the average habitual intake of flavan-3-ol is much lower than that used in most dietary intervention studies [5]. Theaflavins (TFs) are one of the major functional phytochemicals from black and dark tea, forming the unique flavor of special teas and contributing to their health benefits [8,9]. They account for 2–20 g/kg of the dry weight of solids in brewed black tea [10]. The main structure of TFs is a seven-membered benzotropolone ring, followed by decarboxylation and simultaneous fusion with the epicatechin ring B or epicatechin gallate [11]. There are more than 20 identified types of TFs, and theaflavin (TF_1_), theaflavin-3-gallate (TF_2_A), theaflavin-3′-gallate (TF_2_B), and theaflavin-3,3′-digallate (TF_3_) are the major characterized structures [12] (Figure 1).

The antioxidant, lipid-regulatory, anti-inflammatory, antitumor, and antiviral bioactivity of TFs has recently been reviewed [13]. However, the effects and the underlying mechanism of the cell signaling pathway and gut microbiota modulation of TFs on MetS have not been systematically reviewed. The fact that theaflavin has such a multitude of effects gives it great research value and application advancement. Nevertheless, due to its low bioavailability and poor stability, the applications of phytochemicals such as theaflavins are limited [14]. Since the stability and bioavailability of TFs are crucial for their function, they are reviewed in our paper. The effects of pH, digestive enzymes, efflux transport, and metabolism are crucial, and these factors render encapsulation and structure modifications the major methods used for improving the stability and bioavailability of TFs. The purpose of this review is to investigate the functional and regulatory molecular evidence of the response to the treatment of TFs on MetS, with an emphasis on redox homeostasis and gut microbiota balance.

Disciplinary literature (Scopus, Web of Science) and publisher (ScienceDirect, SpringerLink) databases were searched to identify original contemporary full-text articles published on the subject of theaflavin stability, bioavailability, and functionality. Most cited publications were found by using the following search terms: “theaflavin degradation”, “theaflavin thermal stability”, “theaflavin gut bacteria”, “theaflavin stabilization”, “tea polyphenol intestinal microbiome”, “theaflavin obesity”, and “theaflavin diabetes”. The relevance of the given list of publications was further assessed by examining the title and the abstract. Of the total number of reviewed papers, 80% were published in the last 10 years, of which 66% were published in the last 5 years.

## 2. Stability and Bioavailability of Theaflavins

Theaflavins (TFs) in powdered form are quite stable. There is no significantly change in TFs in a simulated tropical environment for 6 months and dry conditions at 65 and 75 °C for 14 days [15]. Heating at a higher temperature (100 °C) for 3 h results in the complete degradation of TFs in aqueous solution, while 56% of which are degraded when heated at 70 °C for 3 h [16]. TF_3_ and TF_2_B are more stable compared with TF_1_ and TF_2_A in boiling water [15]. TFs, as a type of typical polyphenols, exhibit pH-dependent stability. TFs exhibited high stability during 24 h of incubation in sodium acetate buffer at pH 5.5 with simulated gastric juice (0.2% NaCl, 0.24% hydrochloric acid) [17]. On the other hand, TFs are more susceptible to be degraded and form naphthoquinone in alkaline solutions, resulting in the instability of TFs [17]. Under alkaline conditions, TF_3_ and TF_2_B are more stable than TF_1_ and TF_2_A, with pH ranging from 5.0 to 7.4 [15]. At a pH of 7.4, the degradation rate of TFs was 34.8% after incubation for 8 h; when the pH was increased to 8.5, the color of the solution rapidly turned into dark brown, and 78.4% of the TFs were degraded after 2 h of incubation [17]. The instability of TFs regards temperature and pH limiting their application in functional foods and medicinal use.

Bioavailability regards to the proportion of the particular nutrients that are digested, absorbed, and metabolized through normal pathways [18]. TFs are relatively stable during in vitro gastric digestion, which is majorly influenced by pH and digestive enzymes [19]. The effects of pH on the stability of TFs were documented above. In the intestinal phase, TFs exhibited poor bioavailability with apparent permeability coefficient (Papp) values ranging from 0.44 × 10^−7^ to 3.64 × 10^−7^ cm/s [20]. In addition, P-glycoprotein (P-gp), multidrug resistance-associated proteins (MRPs), and breast-cancer resistance proteins (BCRPs) are all involved in the efflux transport of the four TFs, of which P-gp and MRPs have the strongest secretory effect on TFs [20]. Intestinal microbial communities are capable of enzymatically transforming compounds into metabolites, thus potentially affecting their bioavailability [21]. However, the bioavailability of TFs has received limited attention in comparison with that of green tea polyphenols.

Despite some publications having revealed the low bioavailability of TFs, it was surprising that 70% of TFs are more relatively available in the prostate than in EGCG, and a mouse model (mice given a decaffeinated black tea diet) showed that TF_1_ is found in the prostate in conjugated free form [22]. TF_1_ acts as a novel substrate for organic anion-transporting polypeptides (OATPs) 2B1, inhibiting transport and thus reducing pharmaceutical uptake [23]. Therefore, the low bioavailability of TFs can be improved by synergistic effects [24]. Encapsulation methods and structural modifications were used to improve the bioavailability of phenol components [25]. It would be interesting to fabricate emulsions and gel structures to optimize the bioavailability of TFs. Currently, the delivery systems used to encapsulate TFs include emulsifying systems (liposomes, Pickering emulsions) and nanoparticle systems (based on proteins and polysaccharides) [26]. Encapsulation or interaction with polymers including proteins, lipids, and carbohydrates may contribute to improving the stability and bioavailability of TFs. Bovine serum albumin, zein, and liposomes, as carriers for TFs, showed excellent potential to further optimize the bioavailability of nanoparticles [27,28,29]. Hydrophobic interaction and hydrogen bonding are the dominant interactions between TF_3_ and bovine serum albumin, and the microenvironment around bovine serum albumin enhances hydrophobicity with the increase in the α-helical structure, which could potentially be used to develop nanoparticles with excellent biochemical properties [30]. Ding et al. dissolved phospholipid S75, cholesterol, Tween-80, and TF_3_ in a 16:2.4:4:1 mass ratio with ethanol, and combined them with dynamic high-pressure microfluidization to produce nanoliposomes; these nanoliposomes significantly improved the in vitro digestibility of TF_3_ in adverse environments, including weakly alkaline pH and digesting with pancreatin (after 2 h of incubation in simulated intestinal juice, the residual amount of TF_3_ in nanoliposomes and free fluid was 48.42% and 18.24%, respectively) [31]. Srivastava et al. synthesized poly (lactic-coglycolide) nanoparticles (PLGA-NPs), and PLGA-NPs loaded with TF_1_ (encapsulation rate of 18%) showed potential for 7,12-dimethylbenzanthracene (DMBA)-induced DNA repair gene, and potential to inhibit DNA damage-response genes [32]. Nanocomplexes using chitosan (CS) complexed with caseinophosphopeptides (CPPs) via electrostatic interaction encapsulated TF_3_ with high encapsulation efficiency and low cytotoxicity. More importantly, the CS-CPP nanocomplexes significantly improved the intestinal permeability of TF_3_ in a caco-2 monolayer model [33]. The application of encapsulated TFs in the food industry, and their biological fate and mechanism in vivo need to be further explored. The preparation methods of TFs also include solvent extraction isolation, chemical reagent oxidation, and polyphenol oxidase [34], which are not efficient for industrial production [35], leading to the high price of pure TFs.

## 3. Metabolic Syndrome and Theaflavins

### 3.1. Antiobesity and Lipid-Lowering Effects

Obesity is defined as abnormal or excessive fat accumulation that presents a risk to health. It is a multifactorial disease that results from chronic positive energy balance and the excess energy stored in adipose tissue in the form of triglycerides, expanding adipose tissue size and increasing body fat deposits [36]. Abdominal obesity is the most frequently observed component of MetS [37]. The effects of TFs on obesity regard to the regulation of the lipid metabolism, the inhibition of lipogenesis, the promotion of favorable lipolysis, and the β-oxidation and induction of energy dissipation (Figure 2).

TFs are associated with lipolysis through the AMPK–FoxO3A–MnSOD pathway. The transfection of 3T3-L1 adipocytes with a reporter gene constructed from GFP driven by the superoxide dismutase (MnSOD) promoter leads to the upregulation of peroxisome proliferator-activated receptor γ (PPARγ) and PPARα, while it downregulates the expression of the CD36, ACS, and FAS genes. TF_3_ increases CPT-1, LCAD, and HSL transcript levels, and promotes the expression profile of genes that favor lipolysis and β-oxidation, i.e., genes that induce energy-dissipation-related genes and brown-fat-related proteins: mitochondrial uncoupling protein-1 (UCP-1) and UCP-2 [38] (Table 1). The family of UCPs are mitochondrial inner membrane carriers that regulate electrochemical potential and dissipate energy in the form of heat [39]. Among them, UCP-2 is specific to white adipocytes, and functions similarly to brown adipocyte-specific UCP-1. The induction of UCP-1 and UCP-2 in adipose tissue increased systemic energy expenditure and alleviated metabolic disturbances [40].

TFs reduced food intake in mice and prevented obesity by inhibiting lipid uptake in vivo. A Yinghong no. 9 black tea infusion (Y9 BTI) (containing 0.56% TFs) significantly downregulated the expression of liver kinase B1 (LKB1), adenosine monophosphate-activated protein kinase (AMPK), and cell surface death receptor (FAS) with higher expression of AMPK phosphorylation (p-AMPK) and activated the acetyl coenzyme a carboxylase (ACC) [41]. The oral administration of black tea extract (BTE) of mice slightly reduced the levels of certain systemic inflammatory markers (IL-1β, iNOS, and Cox2) in adipose tissue [41]. Xu et al. [42] demonstrated that the oral supplementation of Huang Jinya black tea extract in C57BL/6 mice with HFD-induced obesity decreased the body weight with 14.40% and 18.44% in HJBT150 and HJBT300, respectively. Lipid accumulation comprising total fat mass and adipocyte size was reduced in the WAT of BTE-treated mice. It reduced the corresponding mRNA levels associated with lipogenesis peroxisome proliferator-activated receptor (PPAR) γ, CCAAT/enhancer-binding protein (CEBP) α, fatty acid-binding protein 4 (Fabp4), fatty acid synthase (FAS), acetyl-CoA carboxylase (ACC) 1, and stearoyl-CoA desaturase-1 (SCD1), [42]. Pancreatic lipase (PL) plays an important role in fat metabolism and is an effective target for weight control. Theaflavin-3,3′-digallate, theaflavin-3′-gallate, theaflavin-3 gallate, and theaflavin had inhibitory effects on pancreatic lipase, with IC50 of 1.9, 4.2, 3.0, and >10 μmol/L, respectively [43]. Thus, TFs exhibited antiobesity activity mainly through affecting the appetite, reducing lipogenesis, and promoting lipolysis to facilitate lipid accumulation and control WAT expansion.

Dyslipidemia is highly related with many diseases, such as hyperlipidemia, atherosclerosis, and nonalcoholic fatty liver, which lead to complications that endanger human health [44]. Black tea extract alleviated hyperlipidemia in HFD-induced mice, followed by a reduction in triglyceride (TG), free fatty acid (FFA), and total cholesterol levels [42]. TFs effectively reduced lipid levels in HFD-fed ApoE−/− and C57BL/6J mice treated with TFs with elevated levels of serum TG, TC, and LDL-C, and increased the level of HDL-C [45]. Oxidative stress is linked with the balance of lipid metabolism [46]. TF_1_ reduces ROS and MDA levels, and maintains antioxidant enzyme activity in both in vivo and in vitro experiments. The administration of TF_1_ (5, 10 μmol/L) promoted the activity of antioxidant enzymes (SOD, CAT, GSH-Px), and inhibited the process of atherosclerotic plaque formation and aortic histological alterations [45]. In addition, TF_1_ upregulated the Nrf2/HO-1 signaling pathway in vascular endothelial cells by increased microRNA-24 (miR-24) levels, invoking its activation on Nrf2 [45]. A dose-dependent reduction in lipid droplets and deposition in HepG2 cells was observed through the treatment of TF_3_, and the lipid droplets and deposition were reduced in vivo and in vitro in HepG2 cells [47]. TF_3_ downregulated the levels of SREBP-Ic and FAS, with increased activity of CPT1 and phosphorylation of ACC directly bound to and inhibiting the activation of plasma kinase (PK), further confirming its effect on the stimulation of adenosine monophosphate activated protein kinase (AMPK). Regarding AMPK stimulation and downstream targets, the proposed mechanism for a new target for nonalcoholic steatohepatitis treatment is the TF3–PK-AMPK regulatory axis to alleviate lipid deposition [47]. The mRNA levels of PPARα, carnitine palmitoyltransferase 1α (Cpt1α), L-bifunctional enzyme (Ehhadh), and acyl coenzyme a oxidase (Acox1) were significantly increased with black tea extract in vitro to mediate hepatic fatty acid oxidation, decreased the protein levels of FAS, ACC1, and sterol regulatory element binding protein 1 (Srebpl), and ultimately inhibited reduced fatty acid synthesis [42]. The phosphorylation of serine residues of insulin receptor substrate-1 (IRS1) and PI3K-p85 caused the phosphorylation of Akt, which increased synergistic insulin signaling in the insulin/Akt signaling pathway in the WAT of HJBT300 mice [42]. In addition, TFs also significantly reduced ROS production in steatosis hepatocytes and LPS-stimulated tumor necrosis factor-a (TNF-a) production in RAW264.7 cells [48].

**Table 1 ijms-23-07595-t001:** Effects of theaflavins on diseases caused by metabolic abnormalities (↓ indicates down-regulation, ↑ indicates up-regulation).

Type	Related Disease	Cell Line/Animal Model	Treatment	Effects	References
Hyperlipidemia	Obesity	Mouse 3T3-L1 fibroblast	0, 25, 50 μM TF3, 48 h	↓ FAS expression	[38]
				↓ upregulation of CD36 and ACS	
				↑ gene expression of lipid catabolism and β-oxidation	
				↑ CPT-1L, CAD and HSL transcript levels	
				↑ UCP-1, UCP-2	
				↑ Akt (Ser473)	
				↑ PPARα gene expression	
				↓ PPARγ upregulation	
				↓ phosphorylated FoxO3A	
				↓ inactive FoxO3A protein level	
				↑ MnSOD	
				↑ GFP intensity	
		ICR mice	0.5, 1.0, or 2.0 g/kg Y9 BTI for two weeks,	↓ diet consumption	[41]
				↓ abdominal adipose weight	
				↑ fecal triglyceride	
				↓ lipid absorption	
				↑ Protein intake	
				↑ LKB1 and AMPK	
				↑ FAS	
				↑ phosphorylation of ACC.	
				↓ l IL-1β, iNOS, and Cox-2	
		C57BL/6 mice with HFD-induced obesity	150, 300 mg/kg/day black tea extract for 9 weeks, orally	↓ Body weight	[42]
				↓ food intake and body weight	
				↓ Liver and kidney weight	
				↓ WAT lipid accumulation	
				↓ total WAT mass	
				↓ adipocyte hypertrophy	
				↓ BCAAs and AAAs content	
				↑ PPP metabolites	
				↓ PPARα, Cpt1a, Ehhadhm and Acox1	
				↓ FAS, Acc1 and Srebp1	
				↑ p-Acc1 levels	
				↓ p-Irs1 (Ser 318) and PI3K-p85 levels	
				↑ Akt phosphorylation	
				↑ p-AMPK levels	
				↑ insulin signalling synergistically	
				↑ EDRs	
				↓ phospho-elF2α (Ser52)	
				↓ chol	
				↓ hepatotoxicity	
				↑ mRNA level (WAT lipolysis)	
	fatty liver	HepG2	5 μM TF3, 4 h	↓ SREBP-1c	[47]
				↓ FAS	
				↑ CPT1 activity	
				↑ ACC phosphorylation	
				↓ PK activity	
				↓ hepatic lipid accumulation	
				↓ liver steatosis	
Dyslipidemia	Atherosclerosis	HUVEC (CRL-1730)	5, 10 μmol/L TF1, 2 h	↓ ROS	[45]
				↓ MDA	
				↑ SOD, CAT, and GSH-Px	
				↑ Nrf2	
				↑ down-stream protein HO-1	
				↑ miR-24	
		ApoE^-/-^mice, C57BL/6J mice	5, 10 mg/kg TF for 12 weeks, intragastrically	↓ serum TG, TC, and LDL-C elevation	
				↑ HDL-C	
				↓ vacuoles size and number	
				↓ atherosclerotic lesion area	
				↓ MMP-2	
				↓ MMP-9	
				↓ ROS	
				↓ MDA	
				↑ antioxidant enzymes activities	
Dysglycemia	type 2 diebete	C2C12(T2D)	20 μM TF1, 48 h	↑ Ca^2+^ abundance	[49]
				↑ mitochondrial abundance	
				↑ CaMKK2	
				↑ AMPK	
				↑ PGC-1α	
				↑ SIRT1	
				↑ mitochondrial metabolic activity	
				↑ 2-NBDG uptake	
				↑ total GLUT4	
		HepG2	2.5, 5, 10 µg/mL TFs, 24 h	↑ membrane bound GLUT4	[50]
				↓ IRS-1 (Ser307)	
				↑ Akt (Ser473)	
				↑ glucose uptake	
				↑ insulin sensitivity	
				↑ mtDNA copy number	
				↓ PGC-1β	
				↑ PRC	
				↓ TC uptake	
				↓ blood glucose level	
		HFD-induced mice	TF1, TF2a, TF3 100 mg kg/d, and TFs 200 mg kg/d for 9 weeks	↓ serum glucose	[51]
				↓ TC, TG, LDL and HLD	
				↑ SIRT6 expression	
				↓ SREBP-1 and FASN expression	
				↓ Serum glucose	
				↑ glucose tolerance	
		SDT rats	2 mL theaflavin extract in 0.5% CMC, 25 mg/kg/day for 10-, 16-, 22-, 24- and 28-wk, orally	↑ plasma insulin levels	[52]
				↑ GLP and GLP1	
				↑ incretin secretion	
				the development of pre-diabetes in control, affect glucose transporter expression	
				↓ blood glucose levels	
				↑ plasma insulin	
		streptozotocin-induced diabetic rats	theaflavin (25, 50 and 100 mg/kg b.wt.) in 0.5 mL water for 30 days, intra- gastrically	↓ HOMA-IR index	[53]
				↑ total hemoglobin	
				↓ HbA1C	
				↓ hexose, hexosamine, fucose,	
				and sialic acid in plasma	
				↓ TCA cycle key enzymes activities	
				↑ plasma insulin level	
				↓ TG	
				↓ FFA	
		1 µM alloxan and 4% glucose induced diabetic Zebrafish model.	TF3 (0.5, 2, 4, 6.7, 10, and 20 µg/mL) or metformin hydrochloride (10 µg/mL) for 24 h	↓ glucose level	[54]
				↓ PEPCK level	
				↑ GCK expression	
				↑ β cell regeneration rates	
uric acid metabolism	Hyperuricemia	Kunming male mice of SPF einjected with PO-induced Hyperuricemia	20, 50 and 100 mg/kg/day TF, TF-3-G and TFDG for 7 days, intragastrically	↓ SUA values	[55]
				↓ serum Cr values	
				↓ ADA	
				↓ XOD	
				↓ URAT1	
				↓ GLUT9	
				↑ ABCG2 mRNA	
				↓ OAT1/2	
				↑ OCTN1, OAT1 and OAT2 mRNA	
				↓ inflammatory cells	
				↑ Nrf2 and HO-1	

### 3.2. Hypoglycemic Activity

Type 2 diabetes (T2D) is characterized by relative insulin deficiency caused by pancreatic β-cell dysfunction and insulin resistance in the target organs, contributed by genetic and environmental factors [56]. Sarcopenia may be a cause and consequence of T2D in the aging population, and daily protein intake involving certain amino acids and amino compounds could improve muscle strength, muscle function, and protein synthesis, playing key role in T2D status [57]. Amino acid deprivation was also related to stress response in a HepG2/C3A cell model [58]. TFs act as enzyme inhibitors to slow carbohydrate absorption, controlling starch digestion and regulating postprandial hyperglycemia [59]. TF_2_A had a strong inhibitory effect (92.3% inhibition ratio) on α-glucosidase (α-GC) via competitive inhibition mode, and stable complexes are spontaneously formed by hydrophobic interactions, resulting in a change in the α-GC secondary structure [60]. TFs can inhibit the activity of α-amylase, thereby delaying the digestion of starch. TF_2_B and TF_2_A have stronger inhibitory effects than those of TF_1_, and TF_2_B is a competitive inhibitor, while TF_1_ and TF_2_A are mixed inhibitors with both competitive and noncompetitive inhibitions [61].

Besides the inhibitory effects of enzymes, the suppression of glucose absorption is of great importance in modulating diabetes. TFs might be involved in the regulation of specific components of the PGC family to alleviate diabetes [62]. In a palmitate-induced insulin resistance HepG2 cell model, TFs significantly increased glucose uptake and recovered the mitochondrial function, followed by upregulation of total membrane-bound glucose transporter protein 4 (GLUT4) and phosphorylated Akt (Ser473) levels, and the downregulation of PGC-1β mRNA levels and IRS-1 Ser307 phosphorylation levels [50]. Sodium-dependent glucose transporter 1 (SGLT1) is a transmembrane protein located in the apical membrane od enterocytes, and TF_1_ (40 μM) inhibited the expression of SGLT1 instead of glucose transporter protein 2 (GLUT2), which is a high-capacity transporter [63]. The mechanism of SGLT1 for transporting glucose involves the cotransport of Na+, and depolarizes the plasma membrane as a way to open the Ca^2+^ channels. TFs activated the CaMKK2-AMPK signaling pathway via Ca^2+^ influx and upregulated the expression of PGC-1α and SIRT1 in the C2C12 cell line, thereby promoting myotubular mitochondrial abundance and glucose uptake [49]. A TF mixture (25 mg/kg/day) improved impaired glucose tolerance and significantly lower blood glucose levels in prediabetic SDT rats, and increased insulin expression via the inhibition of gastric inhibitory polypeptide (GIP) and glucagon-like peptide-1 (GLP-1) degradation [52]. Treatment with TFs reduced blood glucose levels and improved insulin resistance in mice, significantly reduced serum total cholesterol (TC), total cholesterol (TG), and low-density lipoprotein (LDL) levels, and inhibited alanine amino transferase (ALT), aspartate amino transferase (AST) activity [51]. The expression levels of sterol regulatory element-binding transcription factor 1 (SREBP-1), fatty acid synthase (FASN), which increases the expression of sirtuin 6 (SIRT6), were increased [51] High-fat diets and streptozotocin-induced diabetic rats were treated with different doses of TFs (25, 50, and 100 mg/kg b.wt/day) for 30 days, which resulted in glycated hemoglobin, hemoglobin, and glycoproteins (hexose, hexosamine, polystyrene and sialic acid), TCA cycle enzymes (isocitrate dehydrogenase, a-ketoglutarate dehydrogenase, succinate dehydrogenase and malate dehydrogenase) returned to near-normal levels, and the decreased homeostatic model assessment of insulin resistance (HOMA-IR) index in a dose-dependent manner [53]. Other pathophysiological mechanisms may explain hyperglycemia. Pancreatic β cells are responsible for producing, storing, and releasing insulin to maintain glucose homeostasis [64]. TF_3_ could restore the size of damaged islets, promote islet β cell proliferation in an alloxan-induced zebrafish model, increase insulin secretion, and regulate blood glucose through a reduction in the levels of phosphoenolpyruvate carboxykinase (PEPCK) and hexokinase isoenzyme (GCK) [54]. Thus, TFs could promote glucose homeostasis and prevent the development of insulin resistance by regulating the IRS-1/Akt/GLUT4 signaling pathway. In addition, TFs act as an enzyme inhibitor, reduces glucose transport activity, and increases mitochondrial abundance, thus delaying glucose transport and absorption in the intestine.

### 3.3. Uric Acid Lowering Effect

Hyperuricemia is an abnormal purine metabolic disease that occurs when uric acid is excessive in the blood, and is significantly correlated with cardiovascular diseases such as coronary heart disease, stroke, and hypertension [65]. TFs can be used as an inhibitor of xanthine oxidase (XO) to improve the antihyperuricemia effect of hyperuricemic mice. TF_1_ is a competitive inhibitor with a significant reversible inhibition of XO (IC50 values, 63.17 + 0.13 μmol/L), and the main driving factors are hydrophobicity and hydrogen bonding [66]. TF_1_ interacts with some residues around an active XO cavity, including Glu-879, Pro-1012, Val-1011, THR-1010, LYS-771, Glu-802, Pro-1076, LEU-873, LEU-1014, ASN-768, LEU-648 and Phi-649 [66]. TFs exerted significant UA-lowering effects on hyperuricemic mice, reducing serum BUN and Cr values while inhibiting ADA and XOD activity to improve renal damage in hyperuricemic mice [55]. TFs downregulated the gene and protein expression of glucose transporter 9 (GLUT9) and urate transporter 1 (URAT1), and dose-dependently upregulated the genes for organic anion transporter-1(OAT1), organic cation transporter n1 (OCTN1), OCT1/2, and OAT2 [55]. Accordingly, the mechanism of TFs in the prevention and treatment of hyperuricemia may be related to inhibiting xanthine oxidase activity, regulating the mRNA and protein expression levels of related anionic transporters.

## 4. The Interactions of Theaflavins and Gut Microbiota

The gut microbiota have been implicated in the pathogenesis of MetS as a constellation risk factor to progress greater metabolic defects [67]. Hypercaloric diets are regarded as the major contributor to the development of the obesity epidemic in the United States and the rest of the world [68]. For example, a high-fat diet usually leads to poor microbiotic health, which leads to the eventual onset of chronic disease [68]. While the gut microbiome is responsive to large swings in food and nutrients [69], sufficient evidence shows that polyphenols can alter the composition of gut microbiota by increasing or decreasing both beneficial and harmful microbes [70]. TFs impact the gut microbiota and contribute to its health-promoting effects [69]. After humans ingest black tea, only trace quantities of TFs are absorbed in the upper gastrointestinal tract [71], indicating that TFs would potentially be subjected to the bacteria-mediated catabolism [72]. Studies support that TFs that reach the large intestine undigested are modified by ring-cleavage, reduction, hydrolysis, decarboxylation, and dihydroxylation reactions [73,74]. TFs could be hydrolyzed by gut microflora such as *Bifidobacteria* and *Lactobacilli*, which convert them into their corresponding metabolites (TF_1_, TF_2_A, TF_2_B, gallic acid, and pyrogallol) [72,75]. Instead, the gallic acid released from TFs is further converted into 3-0 and 4-0 methyl gallate acid, o-benzyl gallate-1-sulfate and o-benzyl gallate-2-sulfate [76]. Furthermore, TFs could be transferred to some smaller phenolic compounds such as 5-(3′,4′-dihydroxyphenyl)-γ-valerolactone and 3-(3′,4′-dihydroxyphenyl) propionic acid [77]. With 200 mg/kg TFs to germ-free (GF) and conventional mice, TFs were transformed into small phenolic metabolites dihydro-and tetrahydro (DF-TF and TH-TF) through the cleavage of the C-ring by intestinal microorganisms in vivo, instead of reducing benzotropolone [78]. In addition, two key colonic metabolites of theaflavin, 3-4′-hydroxyphenylpropionic acid and gallic acid, protect neuronal cells from oxidative-stress-sensitive strains, and TFs might be involved in the neuroprotective effects of MPTP-induced dopaminergic neurodegeneration by increasing tyrosine hydroxylase (TH) and dopamine transporter (DAT) immune responses, and reducing the appearance of caspases in SN [76,79]. TFs also have a regulatory effect on the intestinal microflora. TFs and EGCG have similar flavan-3-ol building blocks, to the extent that they may exhibit similar intestinal microbiota-modulating effects [80]. Microbiomic analysis through 16S rRNA gene sequencing shows that polyphenon E and TFs treatments significantly alter the bacterial community structure in the cecum and colon, but not in the ileum [81]. Particularly, several typical species of probiotic and harmful intestinal microorganisms interact with TFs (Figure 3). A recent study on TF_3_ on the modulation of microbial metabolism during in vitro fecal fermentation showed the growth promotion of *Bacteroides*, *Lachnoclostridium*, *Faecalibacterium*, *Parabacteroides,* and *Bifidobacterium*, while Prevotella and Fusobacterium were significantly inhibited (*p* < 0.05) [77]. TF_3_ has a weaker inhibitory effect on *E. coli* than that of EGCG, and TF_3_ also shows a specificity rise in Dialiser levels [77].

TFs could positively reshape the composition of gut flora [82]. TFs play a beneficial role in regulating leaky and dysregulated intestinal homeostasis by controlling the LMD signaling pathway by blocking the incorporation of LMD, and could delay intestinal epithelial dysfunction, thereby preventing DSS-induced colitis in mice [83]. TF_3_ has the potential to be a broad-spectrum pharmaceutical as an antibacterial agent capable of inhibiting the growth of Gram-positive and -negative, and acid-resistant groups of bacteria with an antisporulating agent (250 ug/mL of TF_3_ inhibited bacterial growth by 99.97%, and 625 ug/mL TF_2_B inhibited spore germination by 99.92%). For example, it inhibited the growth Gram-negative bacteria *Klebsiella aerogenes* (*K. aerogenes*), *Escherichia coli* (*E. coli*), *Pseudomonas aeruginosa* (*P. aeruginosa*), and *Proteus mirabilis* (*P. mirabilis*), and Gram-positive bacteria *Staphylococcus aureus* (*S. aureus*), *Staphylococcus aureus* (*S. aureus*), *Streptococcus pyogenes* (*S. pyogenes*), and *Mycobacterium smegmatis* (*M. smgmatis*) [84,85]. TFs on the pathogenic bacterial population in the oral cavity were also investigated [86]. In addition, Lagha reported that TFs possessed similar activity to that of EGCG, which could reduce the relative abundance of lipopolysaccharide-producing bacteria [87]. Furthermore, by exposing TF_1_ to germ-free (GF) (mice colonized with specific-pathogen-free microbiota) and conventionalized mice, the gut microbiota enhanced the amination and MGO conjugation of TF_1_, which then removed these endogenous metabolic toxins [88]. Interestingly, gallic acyl ester substitution at the 3-position hydroxyl group of the C-ring contributed to bacterial *Escherichia coli* β-glucuronidase (EcGUS) inhibition and effectively alleviated drug-induced gastrointestinal toxicity, as described by Sun et al. [89]. TF-3-G, TF-3′-G, and TFDG inhibited EcGUS more strongly than GCG, ECG, and EGCG did. Thus, TFs are capable of exerting a prebiotic effect on gut microbiota by increasing the abundance of potentially beneficial bacteria (e.g., *Lachnoclostridium* and *Bifidobacterium*) and decreasing the abundance of potentially harmful bacteria (e.g., *Prevotella* and *Faecalibaculum*), and provided enteroprotective benefits by combining toxic metabolites to reduce their toxic effects.

The modulation effects of TFs on MetS through gut microbiota were mainly targeted on the gut barrier, gut–brain axis, and gut–liver/adipose axis, especially by increasing the beneficial gut microbiota and metabolites such as short-chain fatty acids (SCFAs), bile acids (BAs), and amino acids (AAs) [42,90,91,92]. As one significant structural component of the intestinal barrier, gut microbiota play an imperative role in sustaining the intestinal microecosystem, while TFs and their metabolies could reshape microbial profiles and confer protective actions onto the gut barrier [82,93].

The impaired intestinal epithelial barrier function and immune tolerance to intestinal flora in susceptible hosts may be the reasons for a series of intestinal inflammatory diseases such as inflammatory bowel disease (IBD) [94,95]. TFs inhibited neutrophil adhesion, ICAM-1 and VCAM-1 mRNA, and protein expression in LPS-induced RIE cells [96], suggesting that the TFs could be beneficial for the treatment of IBD. The circadian rhythm (CR) of the host and the gut microbes that interfere with each other were suggested to cause various chronic disease problems, such as fatty liver, type 2 diabetes, and chronic gastroenteritis, which affect the metabolism via multi-organ crosstalk (enteric–liver–brain axis etc.) [96,97,98]. A recent study found that TFs significantly modulated the circadian clock oscillations of the intestinal flora and the transcription of circadian clock genes induced by continuous dark (CD) treatment in mice [99]. However, many studies fall short because of the lack of animal models with comparable genetic susceptibilities with the human biology, especially when the role and specific mechanism of TFs in the regulation of human intestinal flora on metabolism remain unrevealed. Thus, clinical trials and the establishment of animal models more closely related to the human intestinal flora are necessary. Moreover, there are very few studies on the effects of pure TFs on MetS by targeting gut microbiota, and further evidence on the gut–brain and gut–liver/adipose axes is needed.

## 5. Conclusions and Future Perspective

As the pathogenesis of metabolic diseases involves multiple pathways, treatment targeting multiple factors is expected to address the driving forces of these diseases’ progression. TFs, which are relatively abundant in fermented teas such as black tea, are advantageous in preventing and treating MetS through modulation on the lipid and glucose metabolism, and the gut microbiota (Figure 3). However, the bioavailability of TFs is quite low, which has led to the further advanced strategies for enhancing TF stability, such as encapsulation and structural modification. Interestingly, the use of TFs targets chronic diseases by influencing multisignaling networks. Thus, more investigations are required to develop an understanding of their beneficial role by multiomic integration, particularly in clinical studies and applications. According to the available evidence, the administration of theaflavins as a nutritional supplement in a well-balanced, nutrient-dense diet or the oral use of theaflavin-enriched tea beverages and loaded tablets may be beneficial for people with chronic diseases. Furthermore, few articles have reported on pure TFs in relation to such aspects, with the majority of studies focusing on black or postfermented teas. Therefore, more indepth studies are required to explore the complex relationships between different TFs and the intestinal flora. The dosage and timing of administration also need to be considered, which might be beneficial for the development of more personalized nutritional and functional foods, and even clinical treatments of TFs.

## Figures and Tables

**Figure 1 ijms-23-07595-f001:**
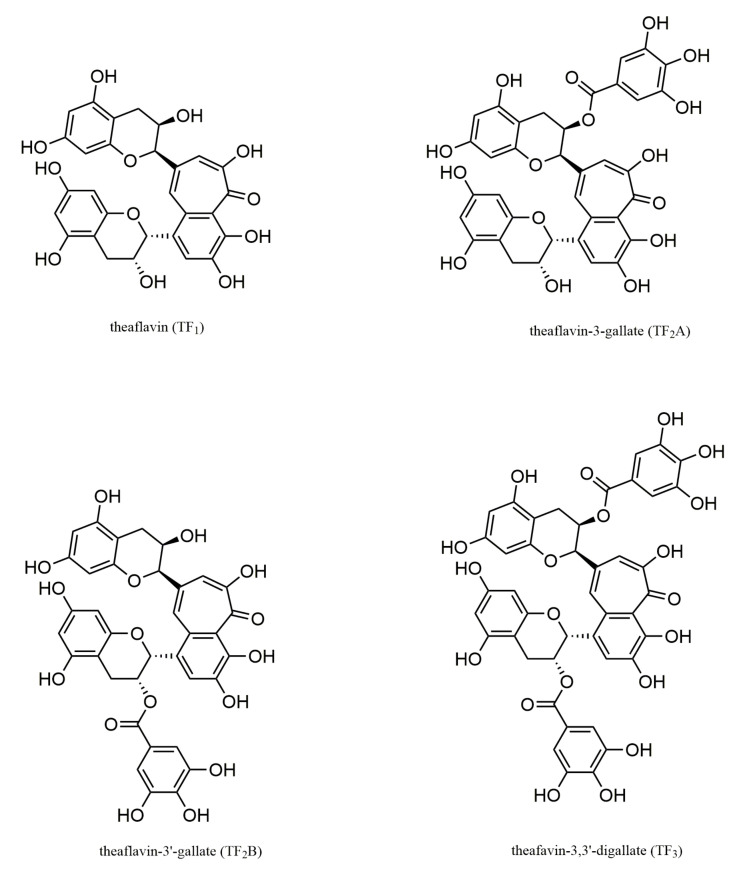
The main monomer forms of theaflavins.

**Figure 2 ijms-23-07595-f002:**
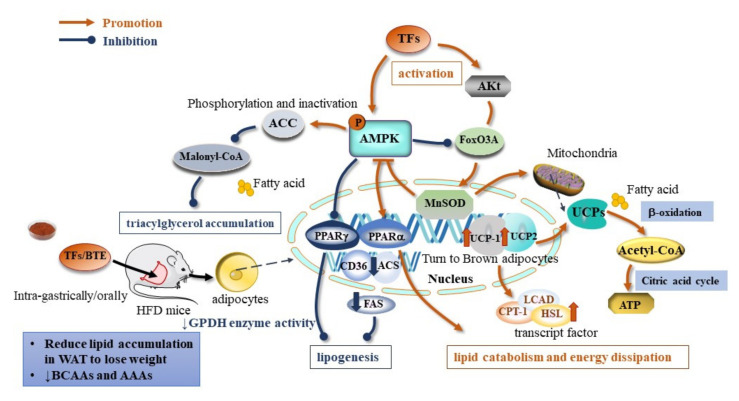
Molecular mechanism of TFs on antiobesity effect (↓ indicates down-regulation).

**Figure 3 ijms-23-07595-f003:**
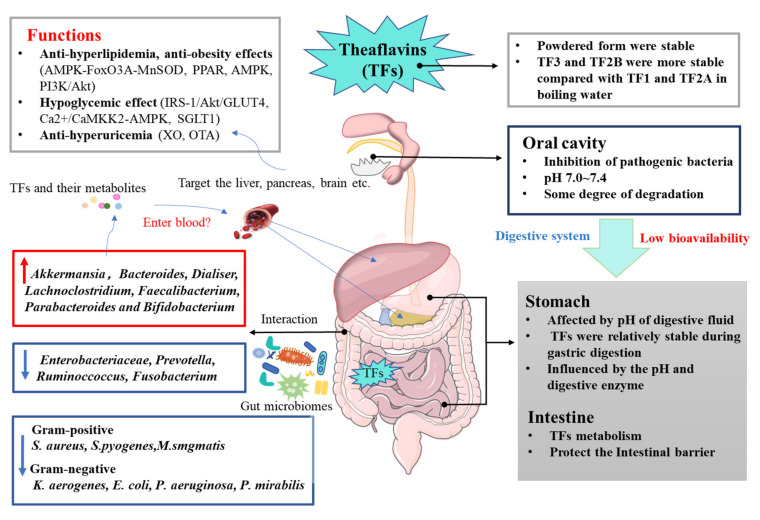
Bioavailability of TFs and mechanism of TFs on metabolic syndrome.

## Data Availability

The data presented in this study are available in the paper.

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
