# Peer review of "Beneficial Effects of Theaflavins on Metabolic Syndrome: From Molecular Evidence to Gut Microbiome"

_ijms, 2022, doi:10.3390/ijms23147595_

Round 1
Reviewer 1 Report
The manuscript submitted for publication by She et al., titled: "Beneficial effects of theaflavins on metabolic syndrome: From molecular evidence to gut microbiome" is an interesting review covering a topic that could have significant clinical implications.
The reviewer would like to offer the following points for the improvement of the manuscript.
1. When discussing metabolic syndrome and type 2 diabetes it is important to underline the importance of diet as a whole in terms of a dietary scheme in addition to the importance of particular nutrients and/or bioactive compounds. Even more so when a discussion involved the gut micro biome. The authors should consider discussing briefly the diet in regards to the gut microbiome and diabetes. A useful paper to consider including is:
- Sikalidis, A.K.; Maykish, A. The Gut Microbiome and Type 2 Diabetes Mellitus: Discussing A Complex Relationship. Biomedicines2020, 8, 8. https://doi.org/10.3390/biomedicines8010008.
2. The authors discuss several potential mechanisms through which improvement of the outcome in terms of metabolic syndrome and diabetes can potentially be achieved. It is important to discuss potentially how these mechanisms compare to macronutrients especially protein/amino acids as there is strong evidence that specifically for metabolic syndrome protein/amino acids may be key in terms of improved outcomes. When considering a whole diet it is important to consider those parameters as well rather than discussing compounds in isolation. A couple of papers that may help towards this end are listed below:
- Maykish, A.; Sikalidis, A.K. Utilization of Hydroxyl-Methyl Butyrate, Leucine, Glutamine and Arginine Supplementation in Nutritional Management of Sarcopenia—Implications and Clinical Considerations for Type 2 Diabetes Mellitus Risk Modulation. J. Pers. Med. 2020, 10, 19. https://doi.org/10.3390/jpm10010019.
- Lee JI, Dominy JE Jr, Sikalidis AK, Hirschberger LL, Wang W, Stipanuk MH. HepG2/C3A cells respond to cysteine deprivation by induction of the amino acid deprivation/integrated stress response pathway. Physiol Genomics. 2008 Apr 22;33(2):218-29. doi: 10.1152/physiolgenomics.00263.2007. Epub 2008 Feb 19. PMID: 18285520.
3. It might be interesting for authors to make a comparison briefly discussing the tea versus wine and/or berries in terms of content of polyphenols and TF in particular. Useful papers to potentially be considered helping in this discussion are:
Nice work overall.
Author Response
Reviewer #1:
The manuscript submitted for publication by She et al., titled: "Beneficial effects of theaflavins on metabolic syndrome: From molecular evidence to gut microbiome" is an interesting review covering a topic that could have significant clinical implications.
The reviewer would like to offer the following points for the improvement of the manuscript.
- When discussing metabolic syndrome and type 2 diabetes it is important to underline the importance of diet as a whole in terms of a dietary scheme in addition to the importance of particular nutrients and/or bioactive compounds. Even more so when a discussion involved the gut micro biome. The authors should consider discussing briefly the diet in regards to the gut microbiome and diabetes. A useful paper to consider including is:
- Sikalidis, A.K.; Maykish, A. The Gut Microbiome and Type 2 Diabetes Mellitus: Discussing A Complex Relationship. Biomedicines2020, 8, 8. https://doi.org/10.3390/biomedicines8010008.
Response: Thank you for your comments. A brief discussion with “Hypercaloric diets were regarded as the major contributor to the development of the obesity epidemic in America and the world [71]. For example, a high-fat diet usually leads to poor microbiota health, which leads to the eventual onset of chronic disease [72].” has been added to line 297 to 300.
- The authors discuss several potential mechanisms through which improvement of the outcome in terms of metabolic syndrome and diabetes can potentially be achieved. It is important to discuss potentially how these mechanisms compare to macronutrients especially protein/amino acids as there is strong evidence that specifically for metabolic syndrome protein/amino acids may be key in terms of improved outcomes. When considering a whole diet is important to consider those parameters as well rather than discussing compounds in isolation. A couple of papers that may help towards this end are listed below:
- Maykish, A.; Sikalidis, A.K. Utilization of Hydroxyl-Methyl Butyrate, Leucine, Glutamine and Arginine Supplementation in Nutritional Management of Sarcopenia—Implications and Clinical Considerations for Type 2 Diabetes Mellitus Risk Modulation. J. Pers. Med. 2020, 10, 19. https://doi.org/10.3390/jpm10010019.
- Lee JI, Dominy JE Jr, Sikalidis AK, Hirschberger LL, Wang W, Stipanuk MH. HepG2/C3A cells respond to cysteine deprivation by induction of the amino acid deprivation/integrated stress response pathway. Physiol Genomics. 2008 Apr 22;33(2):218-29. doi: 10.1152/physiolgenomics.00263.2007. Epub 2008 Feb 19. PMID: 18285520.
Response: Thank you for your references. The content of “Sarcopenia may be a cause and consequence of T2D in the aging population, and daily protein involving certain amino acids and amino compounds could improve muscle strength, muscle function and protein synthesis, playing key role in T2D status [50]. Amino acid deprivation was also related to stress response in HepG2/C3A cell model [51].” has been added to line 225 to 229.
- It might be interesting for authors to make a comparison briefly discussing the tea versus wine and/or berries in terms of content of polyphenols and TF in particular. Useful papers to potentially be considered helping in this discussion are:
Response: Thank you for your comment. The content of “Dietary interventions with flavan-3-ols have shown beneficial effects on metabolic syndrome (MetS). Tea was the major single contributor (75%) to flavan-3-ol monomer intake,followed by pome fruits with 6%, while theaflavins were the main source of the primary flavan-3-ol intake [12]. In countries where tea is consumed sparingly, flavonoid intake relies mainly on proanthocyanidins from berries, contributing to prevent metabolic disorders [13]. The average concentration of flavanols in mg per 100 g fresh weight (mg per 100 ml) where blueberries (58%), red wine (39%) and black tea (28%) [14]. Dietary intakes of flavan-3-ol monomer, proanthocyanidins and theaflavin varied significantly among the countries, and the average habitual intakes of flavan-3-ol were much lower than that of used in most dietary intervention studies [15].” have been added to line 35 to 44 for discussing the tea versus wine and/or berries in terms of content of polyphenols and TF in particular.

Reviewer 2 Report
The review covers the various applications of theaflavins and addresses issues related to their low bioavailability. It is well written and can be accepted for the publication after some revisions
1) Introduction part should be improved by considering all the applications and issues related to their delivery in depth.
2) More detail about nanoscale polymer and lipid drug delivery systems should be given and how they can improve the delivery (possibly in the table).
3) The preferable route of administration should be suggested.
Good luck!!
Author Response
Reviewer #2:
The review covers the various applications of theaflavins and addresses issues related to their low bioavailability. It is well written and can be accepted for the publication after some revisions.
1. Introduction part should be improved by considering all the applications and issues related to their delivery in depth.
Response: Thank you for your comments. The content of “The effects of pH, digestive enzymes, efflux transport, metabolism were crucial, and these factors make encapsulation and structure modifications were the major method used for improving the stability and bioavailability of TFs.” has been added to line 62 to 65 to provide depth information about delivery of TFs.
2. More detail about nanoscale polymer and lipid drug delivery systems should be given and how they can improve the delivery (possibly in the table).
Response: Thank you for your comments. The detailed information about nanoscale polymer and lipid drug delivery systems with “Bovine serum albumin, Zein and TiSiO4, as carriers for TFs, showed excellent potential to further optimize bioavailability of nanoparticles [27-29]. Hydrophobic interaction and hydrogen bonding were the dominate interaction between TF3 and bovine serum albumin, and the microenvironment around bovine serum albumin enhanced hydrophobicity with the increasing of α-helical structure increases, which could be potentially used to develop nanoparticles with excellent biochemical properties [30]. Ding et al. dissolved phospholipids S75, cholesterol, Tween-80, and TFDG in a 16:2.4:4:1 mass ratio with ethanol and combined with dynamic high-pressure microfluidisation to make nanoliposomes, and these nanoliposomes significantly improved the in vitro digestibility of TFDG against adverse environments including weakly alkaline pH and digesting with pancreatin (after 2 h incubation in simulated intestinal juice, the residual amount of TFDG in nanoliposomes and free fluid was 48.42% and 18.24%, respectively) [31]. Srivastava et al. synthesized poly (lactic-co-glycolide) nanoparticles (PLGA-NPs), and PLGA-NPs loaded with TF1 (encapsulation rate of 18%) showed the potential for 7,12-dimethylbenzanthracene (DMBA)-induced DNA repair gene and potential to inhibit DNA damage response genes [32].” has been added to line 118 to 133. Considering the limited investigations about this, it would be better to document this part in manuscript than in a table.
3. The preferable route of administration should be suggested.
Response: Thank you for your comments. “According to the available evidence, the administration of theaflavins as a nutritional supplement in a balanced, nutrient-dense diet, the oral use of theaflavins-enriched tea beverages or loaded tablets may be beneficial for people with chronic disease.”have added to line 400 to 403.

Round 2
Reviewer 1 Report
The authors have provided a reasonable response addressing the reviewer's points. Proofreading is recommended.